

# Testing the knowledge of artificial intelligence chatbots in pharmacology: examples of two groups of drugs

Marcin Mateusz Granat, Aleksandra Paź and Dagmara Mirowska-Guzel

Department of Clinical and Experimental Pharmacology, Medical University of Warsaw, Warsaw, Poland

## ABSTRACT

**Objectives:** The study aimed to evaluate eight artificial intelligence chatbots (ChatGPT-3.5, Microsoft Copilot, Gemini, You.com, Perplexity, Character.ai, Claude 3.5, and ChatRTX) in answering questions related to two pharmacological topics taught during the basic pharmacology curriculum for medical students: antifungal drugs and hypolipidemic drugs.

**Methods:** Chatbots' performance was assessed by answering 60 single-choice questions on antifungal and hypolipidemic drugs topics. The questions were designed to have four answers (a, b, c, and d), and the artificial intelligence (AI) role was to choose the proper one. The assessment was performed twice with a 1-year hiatus to determine if artificial intelligence chatbots' effectiveness changed over time. All the answers were checked for being right or wrong according to up-to-date pharmacology knowledge. To improve the clarity of results, to each score, a mark was assigned based on the grading system applied in our unit. Statistica software version 13.3 and Microsoft Excel 2010 were used for statistical analysis.

**Results:** In 2023, the best results on the subject of antifungal drugs were obtained by Gemini (formerly Bard) and on the topic of hypolipidemic drugs by You.com (formerly YouChat). In 2024 Microsoft Copilot answered correctly the highest number of questions in both topics. The total results of all artificial intelligence chatbots in 2023 and 2024 were compared using t-test for dependent samples. Statistical analysis revealed that artificial intelligence chatbots improved over time in both pharmacological topics, but this change was not statistically significant ($p = 0.784$ for antifungal drugs subject and $p = 0.056$ for hypolipidemic drugs).

**Conclusions:** The accuracy of AI chatbots' responses regarding antifungal and hypolipidemic drugs improved over one year, though not significantly. None of the tested AI systems provided correct answers to all questions within these pharmacological fields.

Corresponding author
Dagmara Mirowska-Guzel,
dagmara.mirowska-guzel@wum.edu.pl

## INTRODUCTION

In 1943 began the construction of the first electronic and programmable computer called the Electronic Numerical Integrator and Computer (ENIAC). At that moment question arose whether machines are intelligent (*Kissinger, Schmidt & Huttenlocher, 2022*). In

1950 mathematician and code breaker Alan Turing proposed an answer to this problem claiming that a machine should be considered "intelligent" if its behaviour cannot be distinguished from that of a human (*Turing, 1950*). This approach was labelled as the Turing test, and while its literal interpretation would allow passing only robots that are identical to humans, the pragmatic approach would rather call "intelligent" all machines that act in a human-like manner. According to that, chatbots can be classified as artificial intelligence (AI) systems based on their ability to mimic human conversation (*Kissinger, Schmidt & Huttenlocher, 2022*; *Tam et al., 2023*).

Chatbot is a computer program and a human-computer interaction (HCI, technology allowing user to interact with a computer *via* a specific interface) model that simulates communication by text or sound with human users, especially over the Internet (*Adamopoulou & Moussiades, 2020*). The AI chatbot's field is under constant development and refinement. Nowadays, this type of AI system already nearly matches or even exceeds human performance in tasks like, *e.g.*, competition-level mathematics, reading comprehension, and image classification (*Jones, 2024*). Chatbots are present in an enormous number of human activities. They are promising tools in medical education and the improvement of patient communication (*Sallam, 2023*). Chatbots can also generate queries that lead to high-precision searches, so their value for researchers who conduct systematic reviews is recognised (*Wang et al., 2023*). They have applications in customer service as an alternative to frequently asked questions (FAQ) by providing detailed replies (*Nirala, Singh & Purani, 2022*; *Kovacevic et al., 2024*).

Finally, chatbots are perceived as practical tools with future potential in treatment in many medical fields (*e.g.*, psychiatry, critical care nephrology, and urology) (*Cheng et al., 2023*; *Suppadungsuk et al., 2023*; *Talyshinskii et al., 2024*). A notable example is DiabeTalk, the AI chatbot with the ability to employ natural language understanding and decision-making algorithms to predict diabetes type and respond to user enquiries (*Rossi et al., 2024*). Another illustrious example of AI utilisation in medicine was presented in research by *De Roberto et al. (2024)*. In this study, ChatGPT-4o was analysing electrocardiogram images with the goal of assisting in the diagnosis of cardiovascular conditions.

As of today, chatbots have the ability to write referenced essays, and in the future, they may replace or be integrated into all search engines (*Stokel-Walker, 2022*, *2023*). Companies that develop AI do not always publish analyses on testing their systems. While the United States Food and Drug Administration (FDA) has approved hundreds of medical devices with AI implemented in healthcare facilities, between the years 2020 and 2022, only 65 randomized controlled trials of AI interventions were published (*Lenharo, 2024*; *Martindale et al., 2024*).

Pharmacology is one of an extremely fast-developing branch of medicine (*Brown et al., 2022*). Thousands of clinical trials are conducted each year worldwide, and huge number of clinical recommendations incorporating new data on drug application appear (*European Medicines Agency, 2024*; *Seoane-Vazquez, Rodriguez-Monguio & Powers, 2024*). Additionally, old drugs gain new indications, and drug repurposing is one of the most popular tools in the modern discovery of new therapies (*Pushpakom et al., 2019*). There is a

need for pharmacological researchers and industry to work together to develop and implement AI technologies, as well as an urgency to benchmark AI performance as an educational tool (*Aziz et al., 2024*; *Shahin et al., 2025*). Because there is limited data about chatbots' performance in the pharmacology field, this study's objective is to provide that evaluation.

## MATERIALS AND METHODS

The aim of the study was to determine the ability of eight chatbots to correctly solve single-choice questions related to pharmacological knowledge on two groups of drugs: antifungal and hypolipidemic. Subsequently, based on the number of correctly answered questions, the comparison of chatbots was conducted. What is more, possessing knowledge that AI systems are constantly changing (*Tebenkov & Prokhorov, 2021*), the aforementioned questions were presented to chatbots twice with a 1-year break. The aim of that was to investigate if any alterations in their performance could be noticed.

In this study, Statistica software version 13.3 and Microsoft Excel 2010 were used for statistical analysis. A 95% confidence interval of the difference was applied for all statistical tests, so $p < 0.05$ was considered statistically significant. Evaluation if data are distributed normally was conducted using Lilliefors test and Shapiro–Wilk test. To assess AI performance before and after 1-year period (from August 2023 to August 2024), a t-test for dependent samples was applied. To highlight performance between AI chatbots analysis of variance (ANOVA) was conducted. For $p$ values with borderline statistical significance threshold Cohen's $d$ value was calculated to address effect size between groups.

### Chatbots selection

AI chatbots were chosen on July 2, 2023, using search engines like Google, DuckDuckGo, and Bing, including phrases: "artificial intelligence chatbot", "artificial intelligence chat", and "AI chatbot". The second search with identical proceeding was performed on July 15, 2024, with the aim of finding new AI chatbots. In this study, we focused on AI systems that represent the large language models (LLMs) as they have already been widely deployed in medicine (*Thirunavukarasu et al., 2023*).

The large language model is an AI model that analyses text present in a vast number of books, articles, and internet-based content. This analysis is possible by using deep neural networks (computing systems inspired by biological neural networks with the ability to perform transformations upon input data), which allows AI to learn specific relationships between words. LLMs chatbots are able to generate responses to given tasks, but their accuracy and coherence are not always correct (*Thirunavukarasu et al., 2023*).

In this study, we focused on free-of-charge AI chatbots accessible *via* the internet and one AI chatbot implemented on GeForce™ RTX 4090 (Nvidia, USA) graphics processing unit (GPU) that we acquired on August 2, 2024.

### Questions and their application

To determine AI chatbots' performance in the pharmacology field, 60 single-choice questions were prepared with four answers marked with letters a, b, c, and d. A total of 30

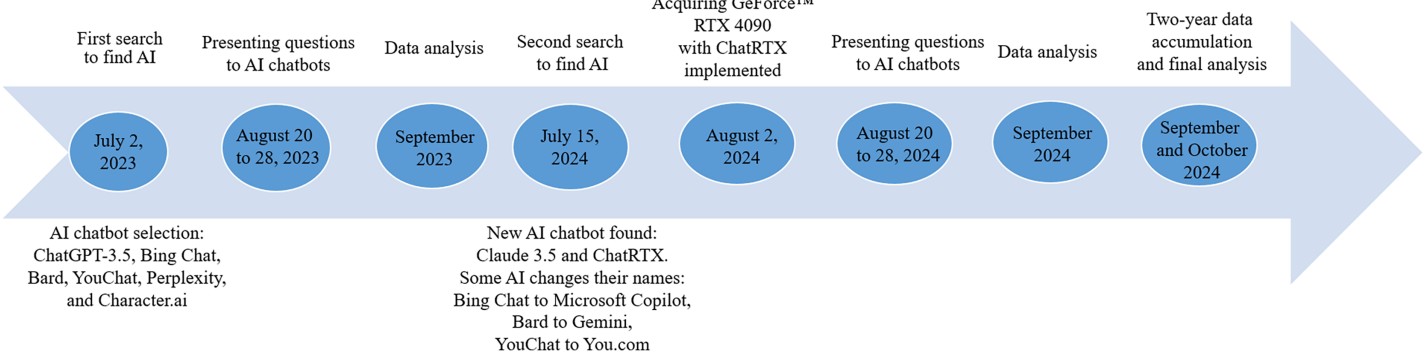

**Figure 1 Scheme presenting steps in testing AI chatbots. AI, artificial intelligence.**

questions were related to antifungal drugs and the other 30 to hypolipidemic drugs. All questions were designed to meet the university level of third-year medical students and were previously used in our recent study, which evaluated the effectiveness of the digital educational game in the pharmacology teaching process (*Granat, Paź & Mirowska-Guzel, 2024*). Three pharmacology experts prepared all questions to ensure the same level of difficulty.

On July 2, 2023, six AI chatbots were found: ChatGPT-3.5 (OpenAI, San Francisco, California, USA), Bing Chat (Microsoft, Redmond, WA, USA), Bard (Google AI, Mountain View, CA, USA), YouChat (You.com, Palo Alto, CA, USA), Perplexity (Perplexity AI, San Francisco, CA, USA), and Character.ai (Character Technologies, Menlo Park, CA, USA). From August 20 to 28, 2023 each chatbot was asked: "I would like to present to you a single-choice test questions about antifungal drugs. Your job will be to choose one correct answer. Is it all right with you?" After the declaration of agreement, 30 questions pertaining to antifungal drugs were presented to each AI chatbot. Subsequently, chatbots were asked a second question: "I would like to present to you a single-choice test questions about hypolipidemic drugs. Your job will be to choose one correct answer. Is it all right with you?", and after consent, 30 single-choice questions about hypolipidemic drugs were asked. All questions were presented one after the other, and in the same order, so chatbots were always working on one task at a time. All the answers were checked for being right or wrong according to current pharmacology knowledge.

On July 15, 2024, two new AI chatbots were found: Claude 3.5 (Anthropic, San Francisco, CA, USA), accessible *via* the internet and ChatRTX (Nvidia, Santa Clara, CA, USA), implemented on NVIDIA GeForce™ RTX 30 or 40 Series GPU or NVIDIA RTX™ Ampere or Ada Generation GPU with at least 8 GB of video random access memory (VRAM) (*NVIDIA, 2025*). At that time, we also noticed that some of the chatbots that we previously used had changed their names. Bing Chat was renamed Microsoft Copilot, Bard became Gemini, and YouChat turned into You.com. ChatGPT-3.5, Perplexity, and Character.ai remained under their original labels. From August 20 to 28, 2024, the same procedure of presenting questions as a year before was employed for all tested AI chatbots (see Fig. 1).

# RESULTS

## First comparison of AI chatbots

First testing took place from August 20 to 28, 2023 and included six AI chatbots: ChatGPT-3.5, Bing Chat, Bard, YouChat, Perplexity, and Character.ai. All AI systems declared that they can answer pharmacological questions. On the subject of antifungal drugs, the best result was achieved by Bard that answered correctly 26 questions out of all 30 (86.7%). This outcome was followed by Bing Chat (24 correct answers, 80%), YouChat (23 correct answers, 76.7%), Character.ai (22 correct answers, 73.3%), Perplexity (20 correct answers, 66.7%), and ChatGPT-3.5 (19 correct answers, 63.3%).

The application of 30 questions related to hypolipidemic drugs concluded that YouChat provided 27 correct answers (90%), which was the highest number among all tested chatbots. Next in line was Bard (26 correct answers, 86.7%), Bing Chat and ChatGPT-3.5 (both with 23 correct answers, 76.7%), Perplexity (22 correct answers, 73.3%), and Character.ai (21 correct answers, 70%).

The analysis of the answers on the subject of antifungal drugs revealed that all six AI chatbots answered correctly questions nos. 1, 2, 3, 8, 11, 16, 17, 18, 19, 21, 22, and 25. Questions nos. 5 and 9 were answered correctly by two AIs; one AI was right in question no. 28, and none provided proper solution in questions nos. 6 and 24. On the subject of hypolipidemic drugs, all six AI systems properly answered questions nos. 3, 6, 8, 10, 11, 12, 13, 16, 20, 21, 22, 23, and 24. Two AI chatbots answered correctly question no. 18 and one AI was right in questions nos. 19 and 29.

## Second comparison of AI chatbots

After a hiatus of 1 year, from August 20 and 28, 2024, eight AI chatbots were employed for analysis. While ChatGPT-3.5, Microsoft Copilot, Gemini, You.com, Perplexity, Character.ai, and Claude 3.5 declared that would answer the questions, ChatRTX stated that it cannot provide any answers related to drugs or medical treatments. On this account, ChatRTX was excluded from this research.

The best result regarding antifungal drugs was achieved by Microsoft Copilot with 27 correct answers to 30 questions (90%). It was followed by Claude 3.5 (26 correct answers, 86.7%), You.com (25 correct answers, 83.3%), ChatGPT-3.5 and Perplexity (both with 23 correct answers, 76.7%), Character.ai (20 correct answers, 66.7%), and Gemini (19 correct answers, 63.3%). To improve the clarity of results, to each score, a mark was assigned based on the grading system applied in our university in which: a failing grade (2) was received when a number of correctly answered questions $(x) \in \langle 0, 15 \rangle$, a satisfactory grade (3) when $x \in \langle 16, 18 \rangle$, a satisfactory plus grade (3.5) when $x \in \langle 19, 21 \rangle$, a good grade (4) when $x \in \langle 22, 24 \rangle$, a good plus grade (4.5) when $x \in \langle 25, 27 \rangle$, and an excellent grade (5) when $x \in \langle 28, 30 \rangle$ (see Fig. 2).

Similarly, in the case of hypolipidemic drugs, Microsoft Copilot was also the best by answering 29 questions correctly (96.7%). Next were: Claude 3.5, Perplexity, and You.com,

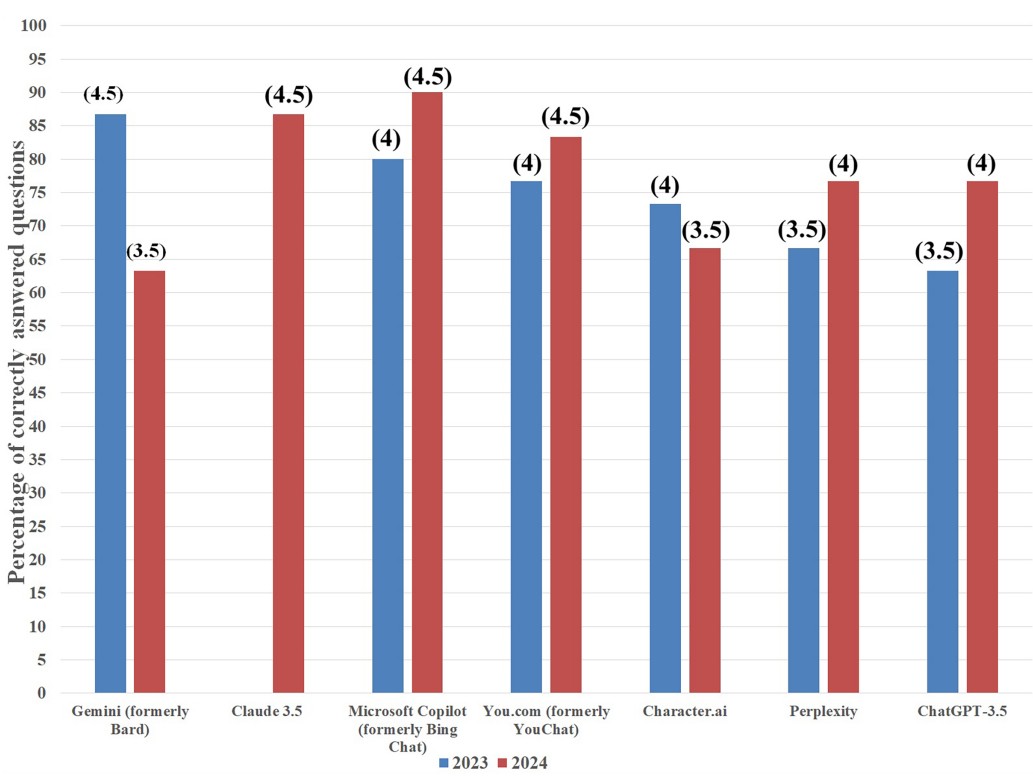

**Figure 2 Artificial intelligence chatbots' percentage of correct answers on antifungal drugs topic in 2023 (blue columns) and 2024 (red columns).** Grades attributed to each score are presented in parentheses.

and all three successfully answered 28 questions (93.3%). ChatGPT-3.5 was next in line with 27 correct answers (90%), and after it were Character.ai (25 correct answers, 83.3%) and Gemini (24 correct answers, 80%). Identically, as with the results obtained in 2023, to each score the grade was assigned (see Fig. 3 and Table 1).

The answers on the subject of antifungal drugs were analysed, which revealed that questions nos. 2, 3, 4, 7, 8, 11, 13, 14, 16, 18, 19, 22, 25, 26, and 27 were properly answered by all seven AI chatbots. Questions nos. 15 and 24 were correctly resolved by two AIs, and questions nos. 5, 6, and 28 were properly solved by one AI. Analogical analysis on answers related to hypolipidemic drugs provided information that all seven AI systems were right in questions nos. 2, 3, 6, 8, 9, 10, 11, 12, 13, 16, 17, 21, 22, 23, 24, 25, 26, 27, 28, and 30. Only one AI answered correctly question no. 29.

## Comparison of total results on AI chatbots performance

As all data obtained in this study represent normal distribution, the calculation of median or arithmetic mean was decided to be sufficient in the preliminary analysis of the results (*Gandhi et al., 2023*). In this study, we choose the median as a worth calculating value. According to that, the performance of all AI chatbots employed in the 2023 analysis characterised by median equalled 22.5 on the subject of antifungal drugs and 23.0 in relation to hypolipidemic drugs. The same calculation applied to all AI chatbots utilised in

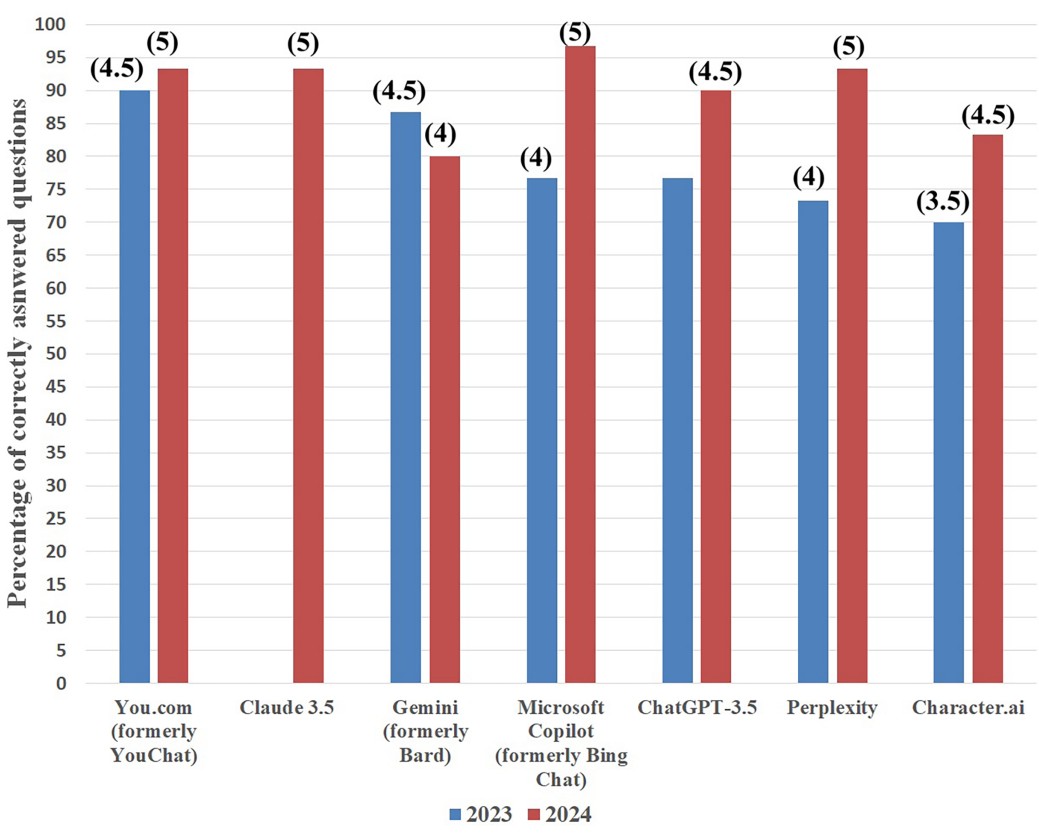

**Figure 3** **Artificial intelligence chatbots' percentage of correct answers on hypolipidemic drugs topic in 2023 (blue columns) and 2024 (red columns).** Grades attributed to each score are presented in parentheses.                                                                         

**Table 1 Artificial intelligence chatbots' accuracy.**

| Artificial intelligence chatbot | Accuracy on antifungal topic | | Accuracy on hypolipidemic topic | |
|---|---|---|---|---|
| | 2023 | 2024 | 2023 | 2024 |
| Gemini (formerly Bard) | 86.7% | 63.3% | 86.7% | 80% |
| Claude 3.5 | – | 86.7% | – | 93.3% |
| Microsoft Copilot (formerly Bing Chat) | 80% | 90% | 76.7% | 96.7% |
| You.com (formerly You Chat) | 76.7% | 83.3% | 90% | 93.3% |
| Character.ai | 73.3% | 66.7% | 70% | 83.3% |
| Perplexity | 66.7% | 76.7% | 73.3% | 93.3% |
| ChatGPT-3.5 | 63.3% | 76.7% | 76.7% | 90% |

the 2024 analysis resulted in a median equalled 23.0 on the topic of antifungal drugs and 28.0 on the subject of hypolipidemic drugs.

Statistical analysis was conducted to determine if AI improvement over a year is statistically significant. Data sets distribution was analysed with the use of Lilliefors test and Shapiro–Wilk test and resulted in conclusion that all the data used in this study have normal distribution.

T-test for dependent samples conducted for antifungal drugs topic resulted in the conclusion that AI chatbots improved their performance, but this change was not statistically significant ($p = 0.784$). Likewise, the t-test for dependent samples employed to analyse data on the subject of hypolipidemic drugs stated that the AI betterment in answering questions was present but was not statistically significant ($p = 0.056$). Analysis of variance (ANOVA) on antifungal and hypolipidemic drugs subjects also resulted in no statistically significant changes with $p$ values, respectively, $p = 0.183$ and $p = 0.860$. Cohen's $d$ value for hypolipidemic drugs equalled 1.61.

All data on antifungal and hypolipidemic drugs together were analysed in relation to 2023 and 2024. The data represented a normal distribution, with median equalled 23.0 in 2023 and 25.5 in 2024. A t-test for dependent samples concluded that improvement of AI chatbots was not statistically significant ($p = 0.124$). Interestingly, all questions applied in this work were previously used in our different study on human participants (*Granat, Paź & Mirowska-Guzel, 2024*). Because of that AI systems results can be preliminarily compared with mean scores of third-year medical students ($n = 66$), which equalled 16.3 on antifungal drugs topic and 16.8 on hypolipidemic drugs topic.

## DISCUSSION

All seven AI chatbots correctly answered most of the questions concerning antifungal and hypolipidemic drugs. Albeit the number of correct answers varied depending on the AI, none of them was inerrable. This is the reason why we conclude that the process of gaining knowledge in pharmacology field is still more beneficial using scientific sources rather than depending completely on AI systems.

Albeit all prepared questions were designed to represent the same level of difficulty, we observed that some of them were correctly answered by all AI chatbots, and others by only two, one, or no AI. No specific patterns (*e.g.*, questions categories) explaining AI performance were noticed. We cannot be certain, but AI difficulties may be a result of poor training data quality which led to inaccurate predictions of some answers (*Stanton, 2025*).

It is apparent that only one AI chatbot, Gemini (formerly Bard), has worsened its scores over time in both pharmacological topics. Although drops in AI performance and behaviour drifts (*e.g.*, changes in following user instructions) have been reported in the past, the causes behind them remain unclear (*Chen, Zaharia & Zou, 2024*). Companies developing AI systems are constantly updating their models, but specifics behind this process remain confidential. Because of that, the end user can only assess results of the updates, but is unable to identify a specific cause or causes of these results at *e.g.*, coding level. Although certainty eludes us, some propositions may suggest explanation. One of them is related to updates. They goal is to improve AI, but due to profound changes, they may result in the opposite effect (*OpenAI, 2023*). This situation may have happened with Gemini that received a major update on 8 February 2024 (*Gemini, 2024*), which was between first and second comparison of AI chatbots in this study. It is possible that these changes were caused by model collapse, which is a degenerative process in which data generated by AI end up polluting the training set of the next generation of generative

model. In effect AI misapprehends reality because is trained on a polluted collection of data (*Shumailov et al., 2024*).

It is important to emphasise that comparison of results on AI chatbots' performance on hypolipidemic drugs resulted in a borderline statistical significance threshold ($p = 0.056$) and Cohen's $d$ value equalled 1.61, indicating that the effect size is large (*Cohen, 1988*). While it is true that AIs improved on the hypolipidemic drugs topic (median equalled 23.0 in 2023 and 28.0 in 2024), one must note that these results are based on aggregated data of seven AIs, whereby practical use of these results for a potential user is limited. It is so, because for an individual the most valuable information is which specific AI represents the best performance.

It is worth noting that the answers we obtained from AI chatbots differed from those of humans. The vast volume of text was generated within seconds in each answer, and presented language was grammatically accurate. It was in opposition to human responses, which are often concise, full of poor vocabulary, typographic errors, and abbreviations (*Hill, Ford & Farreras, 2015*). In our study, we also noticed that AI systems never used emojis, which are popular in human messages. It is evident that in our study a specific human-AI interaction was formed. Although our goal was not to determine the aforementioned interaction, future studies may take that under consideration and provide information on the dynamics of human cooperation with AI, which can be detected using *e.g.*, deep learning (*Freire-Obregón et al., 2020*).

The idea of testing AI systems, whether they can pass medical tests, has already been present in published studies. *Kung et al. (2023)* designed a study in which AI tried to pass the United States Medical Licensing Exam that was consisted of three standardized tests of expert-level knowledge. While AI employed in this research performed at or near the passing threshold of 60%, it is worth mentioning that only one type of AI, ChatGPT, was tested. Another approach, but also related to ChatGPT exclusively, was published by *Peng et al. (2024)*. In this work researchers did not use medical test but applied 131 valid questions from a medical book concerning colorectal cancer. The results stated that ChatGPT did not meet the standards of an expert level.

The assessment of medical knowledge performance related to more than one AI chatbot was conducted by *Pan et al. (2023)*. In this work, a queries related to the five most common cancers were presented to four AI systems, which resulted in the conclusion that the quality of text responses was good and no misinformation was present. A similar study was published by *Mohammad-Rahimi et al. (2024)*. Its methodology was based on an analysis of responses provided by six AI chatbots to questions related to oral pathology, oral medicine, and oral radiology with the use of a 5-point Likert scale. While the highest mean score for performance across all disciplines was 4.066 ± 0.825, the authors additionally evaluated the authenticity of citations generated by AI systems. Interestingly, 82 out of 349 (23.50%) citations were fake, which draws attention to a phenomenon called artificial hallucination—a situation when AI generates sensory experiences that appear real but are fictitious (*Tangsrivimol et al., 2025*). Although it was outside the scope of our study, future research may benefit from determining not only AI chatbots' effectiveness but also the level of their artificial hallucination.

As AI chatbots are becoming more and more popular in medical education, new fields of medicine become present in AI evaluation processes *e.g.*, family medicine in *Hanna et al. (2024)* study. In this work, researchers inputted 193 multiple-choice questions from the family medicine in-training exam written by the American Board of Family Medicine. Three AI chatbots were tested, and the best one scored 167/193 (86.5%), which was higher than residents' mean of 68.4%. Multiple-choice questions were also used in the testing of three AI systems on the subject of postgraduate-level orthopaedics (*Vaishya et al., 2024*). The AI with the highest score had 100% efficiency, but the study did not present information about the results obtained in humans on the same questions.

There is little data on the comparison of AI chatbots performance at answering medical questions in different time intervals. During our research, we found only one study of this type by *Mihalache, Popovic & Muni (2023)*, and it assessed ophthalmic knowledge of ChatGPT from January 9 to 16, 2023, and on February 17, 2023. The research concluded that ChatGPT answered half of the questions correctly, but its limitations were employing only one AI chatbot as well as short hiatus between AI assessments. Studies determining AI systems' performance regarding pharmacology knowledge alone are also very limited. We found one research by *Elango et al. (2023)* who tested ChatGPT abilities in pharmacology examination of phase II Bachelor of Medicine, Bachelor of Surgery (MBBS, a medical degree granted by universities in countries that adhere to the United Kingdom's higher education tradition) with the average total score results of 76%.

To the best of our knowledge, this study is the first assessment of numerous AI chatbots' performance in the pharmacology field with their comparison at two time intervals, applying a 1-year hiatus between them. Certainly, our work has its weaknesses. It is limited to only two pharmacological topics, so it would be beneficial to conduct future analysis with a broader scope of knowledge to enhance the generalizability of the findings. It is possible that AI chatbots' abilities will advance or regress in the course of time. With new versions of tested AI systems, examining their future performance would be worth considering for future research. What is more, as a field of artificial intelligence is steadily developing, it would be valuable to search for potential new AI systems and provide results on their effectiveness. It is worth noting that AI systems are exposed to data biases that may lead to skewed or unfair outcomes (*Hasanzadeh et al., 2025*). We would like to point out that presenting the same set of questions in the same order might introduce pattern recognition benefits for AI chatbots, so it may be beneficial for future studies to consider randomizing question order. What is more, AI chatbots can be trained on question-answer datasets that resemble the test format, which may affect AI responses (*Prakash et al., 2025*).

During all our work, we communicated with AI chatbots exclusively in English, but as large language models (LLMs) can learn other languages, future assessments of AI performance in different languages may be interesting to investigate. Moreover, additional insights on AI chatbots' performance in comparison to humans' abilities in executing the same tasks, would also be of great relevance and may be provided by prospective studies.

## CONCLUSION

The novelty of this study includes a comparison of multiple AI chatbots' performance exclusively in pharmacology over a 1-year period. Moreover, some of the AIs (You.com, Perplexity, Character.ai, and ChatRTX) were never tested in the field of medicine knowledge, let alone pharmacology. The methodology of this work allowed us to determine AI chatbots' effectiveness in two pharmacologic topics, which is a useful insight as AI may be used as an educational tool.

The evaluation of eight AI chatbots resulted in the conclusion that in 2023, six of them answered correctly more than half if the questions (the range of correct answers for the antifungal drugs topic was from 19 to 26, and for the hypolipidemic drugs subject from 21 to 27). Better, but not with statistically significant change, were the results obtained in 2024 when seven AI chatbots were analysed. The range of correct answers for the antifungal drugs subject spanned from 19 to 27, and for the hypolipidemic drugs topic from 24 to 29. It is noticeable that only one AI chatbot, Gemini (formerly Bard), performed worse after a 1-year hiatus in both topics. What is more, ChatRTX was the only AI excluded from this study by being unable to answer enquiries.

As many AI chatbots are available free-of-charge, accessing them is instant and effortless. With rising number of users, it seems appropriate to seek data on AI chatbots' accuracy in delivered answers. While companies developing AI chatbots are not providing transparency tests (*Stokel-Walker, 2023*), determining AI knowledge by *e.g.*, employing methods presented in this study, may be the only tool suitable for verifying AI chatbots' usefulness as a source of information.

One may presume that the performance of AI chatbots in the field of pharmacology will improve over time but such an assumption cannot be fully justified. In our study, one of the evaluated AI chatbots surprisingly worsened within 1 year, whereas others improved, but none had reached a maximal score. It is essential information for students and teachers but also health professionals. They should be vigilant as the results of our study show that depending on AI chatbots may be deceptive.

## ABBREVIATIONS

| | |
|---|---|
| **AI** | artificial intelligence |
| **ENIAC** | Electronic Numerical Integrator and Computer |
| **FAQ** | frequently asked questions |
| **FDA** | Food and Drug Administration |
| **GPU** | graphics processing unit |
| **HCI** | Human-computer Interaction |
| **LLMs** | large language models |
| **MBBS** | Bachelor of Medicine, Bachelor of Surgery |
| **VRAM** | video random access memory |

### Funding

The authors received no funding for this work.

### Competing Interests

The authors report there are no competing interests to declare.

### Author Contributions

- Marcin Mateusz Granat conceived and designed the experiments, performed the experiments, analyzed the data, performed the computation work, prepared figures and/or tables, authored or reviewed drafts of the article, and approved the final draft.
- Aleksandra Paź analyzed the data, performed the computation work, authored or reviewed drafts of the article, and approved the final draft.
- Dagmara Mirowska-Guzel analyzed the data, prepared figures and/or tables, authored or reviewed drafts of the article, and approved the final draft.

### Data Availability

The data is available in the Supplemental Files and Zenodo: Granat, M. M. (2025). Testing the knowledge of artificial intelligence chatbots in pharmacology Examples of two groups of drugs. Zenodo. https://doi.org/10.5281/zenodo.15285002.

### Supplemental Information

Supplemental information for this article can be found online at http://dx.doi.org/10.7717/peerj-cs.2954#supplemental-information.

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
