# Peer review of "Testing the knowledge of artificial intelligence chatbots in pharmacology: examples of two groups of drugs"

_PeerJ Computer Science, doi:10.7717/peerj-cs.2954_

## Round 0.1 · original submission · Minor Revisions

Dear Authors,

I carefully read the reviewer’s reports and concluded that this contribution may be eligible for publication. However, it can benefit from a review, addressing reported concerns.

Reviewer 2 suggested proofing checking language and strengthening the literature review section. Also, the reviewer suggests extending a comparison between proposed and existing methodologies.

Reviewer 1 stated that the manuscript is clear and well-structured. The reviewer suggests extending the literature review using recent studies from 2025 and 2024 and considering other chatbots like, Gemma, DeepSeek, etc…

Also, reviewer 3 comments confirm the need to consider other chatbots in the experimentation, in particular, due to the recent improvements in the fields.
Reviewer 4 provided a very comprehensive list of suggestions that can help in improving the quality of the dissertation and increasing the robustness of your findings.

**Language Note:** The review process has identified that the English language must be improved. PeerJ can provide language editing services - please contact us at [email protected] for pricing (be sure to provide your manuscript number and title). Alternatively, you should make your own arrangements to improve the language quality and provide details in your response letter. – PeerJ Staff

Reviewer 1 ·

Basic reporting

The paper is clearly written and well-structured, with appropriate use of technical language and a coherent narrative throughout. The abstract provides a clear summary of the study's objectives, methods, and conclusions. The introduction effectively frames the importance of testing AI chatbots in the context of medical education and patient care, and it is supported by relevant references from recent literature.

The paper’s figures are relevant, high-quality, and well-labeled, enhancing the understanding of the data presented. The raw data supporting the conclusions has been made available through open-access repositories, which aligns with good research practices.

Suggestions for improvement:

The literature review could include more recent studies from 2024-2025 on AI performance in healthcare, ensuring the references are up-to-date.
A minor language review would help refine sentence clarity and flow in some sections of the discussion.

Experimental design

The experimental design is sound, with a clear research question aimed at evaluating the performance of various AI chatbots across two pharmacological topics. The selection of chatbots and the decision to reassess their performance after a one-year hiatus adds depth to the study.

The methodology is well-described, allowing for replication. The choice of using single-choice questions mirrors real-world testing conditions for medical students, making the study relevant for educational applications.

Suggestions for improvement:

Including a wider variety of pharmacological topics would enhance the generalizability of the findings.
More detailed information on the difficulty level of questions or their validation by external pharmacology experts would strengthen the methodology.

Validity of the findings

The findings are valid and well-supported by statistical analysis. The use of the t-test for dependent samples is appropriate for comparing chatbot performance over time. The results are clearly presented, with sufficient explanation of the statistical significance (or lack thereof).

The conclusions drawn are consistent with the results, acknowledging the limited improvement in chatbot performance over time. However, the discussion could be expanded to consider external factors that might have influenced the performance of AI systems, such as model updates or changes in training data.

Suggestions for improvement:

A more thorough analysis of why some chatbots, like Gemini, performed worse over time would add valuable insight.
Exploring whether certain types of questions were more challenging for the AI models could reveal patterns in their limitations.

Additional comments

The study offers a valuable contribution to understanding how AI chatbots handle specialized medical knowledge and whether their performance evolves over time. It is particularly relevant given the growing use of AI in education and healthcare.

The discussion section effectively situates the findings within the broader context of AI development and its limitations in medical education. The comparison to previous research is well-articulated, and the paper offers meaningful suggestions for future research directions.

Strengths:

Novel longitudinal design evaluating AI chatbot knowledge over time.
Clear, replicable methodology and transparent data availability.
Strong relevance to medical education and the growing role of AI in healthcare.
Weaknesses:

Limited to only two pharmacological topics.
The analysis could be expanded to explore specific question categories or difficulties faced by the chatbots.


Also I suggest to cite: 10.1016/j.cviu.2020.102991
The paper introduces a deep learning model to detect human cooperative behaviors, useful if your paper discusses the interaction between human users and AI-based systems. It can support the discussion on the dynamics of attention and cooperation between humans and artificial intelligence.

·

Basic reporting

The manuscript is very relevant and highly interesting in today's context. Minor revision is suggested to further enhance the quality of this manuscript.

Experimental design

Careful editing of language is essential. Strengthening of literature review section is essential.

Validity of the findings

I think the manuscript could compare with past methods by model architecture and parameters to verify the experimetal result.

Reviewer 3 ·

Basic reporting

The study evaluated the performance of eight AI chatbots (ChatGPT-3.5, Microsoft Copilot, Gemini, You.com, Perplexity, Character.ai, Claude 3.5, and ChatRTX) in answering questions on two pharmacological topics taught in medical school: antifungal and hypolipidemic drugs. Each chatbot answered 30 single-choice questions per topic, with the test repeated after one year to assess improvement over time.

Major comments:
- Please, update reference in the introduction;
- I would suggest the authors to cite for the chatbots used in the state of the art for the different tasks, the following papers: https://doi.org/10.1109/BIBM62325.2024.10821742; 10.1109/BIBM62325.2024.10822822.

Experimental design

No comments.

Validity of the findings

The novelty it is not clear in the manuscript.

- I would suggest the authors to introduce tables about the different performance metrics such as Accuracy, Specificity, sensitivity, and so on;
- There are other recent versions of the chatbot that can be tested. I would suggest the authors, if it is possible, to integrate their works with this new versions;
- I would suggest the authors to introduce their novelty in the contribution;
- The discussion and conclusion should be improved.

Additional comments

No comments.

·

Basic reporting

The manuscript is clearly written, with professional language and a logical structure. The background provides relevant context on AI and its potential in healthcare and education. Figures are well-labeled and visually support the data, and references are generally appropriate.

However, a few areas need improvement:
• The language and grammar require some polishing. For instance, several awkward or unclear phrasings (e.g., lines 36–38: “In 1943 the first electronic and programmable computer called Electronic Numerical Integrator and Computer (ENIAC) was created and in this moment question if machines are intelligent arisen”) should be revised for clarity.
• Figure captions could benefit from more detailed explanations, especially Figures 2 and 3, which lack contextual interpretation in the captions.
• Some acronyms (e.g., MBBS, HCI) are introduced without immediate definition.
• The abstract should more explicitly state the methodological framework (e.g., how scoring was conducted, grading thresholds, chatbot response limitations).

Suggested improvements:
• Perform a thorough grammar and syntax review, possibly with professional editing support.
• Expand figure captions to make them self-explanatory.
• Improve clarity and flow in the Introduction (e.g., condense historical context, sharpen the rationale for pharmacology focus).

Experimental design

The design is generally sound and appropriate for a descriptive benchmarking study.

Strengths:
• Reproducible and transparent methodology: question sets, chatbot interaction procedure, and evaluation framework are well described.
• Use of statistical tests (t-test for dependent samples) and normality checks (Lilliefors, Shapiro-Wilk) aligns with standards.

Areas for improvement:
• The methodology for grading chatbot responses is overly simplified. Assigning grades based on fixed ranges (e.g., “22–24 correct answers = good”) might not be informative without clearer justification or reference to standard assessment criteria.
• There is no human benchmark for comparison (e.g., average score of students), which would give context to chatbot performance.
• Bias control is limited. Asking the same set of questions in the same order might introduce pattern recognition benefits for LLMs trained on public data.

Suggested improvements:
• Justify or refine the grading scale; possibly compare chatbot scores to human student performance (if available).
• Discuss and mitigate potential biases in repeated question presentation.
• Consider anonymizing and randomizing question order in future studies.

Validity of the findings

The results are logically presented and supported by the data. The finding—that chatbot performance improved over time but not significantly—is consistent with the statistical analysis.

However:
• The statistical significance threshold (e.g., p = 0.056 for hypolipidemics) is borderline; discussion of practical vs. statistical significance could be expanded.
• There’s no analysis of variance between chatbots, which would be useful to understand how performance differs across models, rather than grouping all together.

Suggested improvements:
• Expand discussion on effect size and practical implications, especially given the p-values close to significance thresholds.
• Include inter-chatbot variance analysis or ANOVA to highlight performance consistency or variability.
• Acknowledge that some chatbots may have been trained on question-answer datasets that resemble the test format.

Additional comments

This is a timely and well-structured exploratory study addressing an important gap in evaluating AI systems for medical knowledge tasks. The use of two pharmacology topics provides useful insights, and the one-year gap between assessments is a valuable design element.

Still, the narrow focus on only two drug classes limits generalizability. Future work should consider a broader range of topics and include human comparators. Also, qualitative analysis of incorrect answers (e.g., hallucinations or plausible but wrong responses) would enrich the findings.

---

## Round 0.2 · accepted · Accept

Dear Authors,

The reviewers agree that you have addressed all of their concerns.

Reviewer 3 ·

Basic reporting

The paper is now well-written.
The literature references is adequate.

Experimental design

The experiments are well designed.

Validity of the findings

The results achieved are robust.

·

Basic reporting

The authors have directly addressed all major reviewer suggestions in a meaningful and well-integrated manner. The revised manuscript reflects careful attention to both content and presentation. The language has been polished, references have been updated with recent 2024–2025 literature, and acronyms are now clearly defined. Figures and tables have been improved with appropriate captions and accuracy metrics. The manuscript is now clear, well-structured, and aligns with PeerJ’s standards for basic reporting.

Experimental design

The study design is clear, and the methodology has been significantly improved in response to reviewer feedback. The addition of expert validation for question difficulty, human benchmark scores, and ANOVA analysis strengthens the experimental rigor. However, a few limitations remain inherent to the study design, which are acknowledged by the authors—for instance, the narrow focus on two drug classes and the use of fixed question order. Future studies would benefit from randomizing question order and incorporating a broader range of pharmacological topics. These have been appropriately proposed in the discussion, and no further revision is necessary at this time.

Validity of the findings

The authors have addressed concerns related to statistical significance and performance metrics. The inclusion of Cohen’s d effect size and discussion of practical vs. statistical significance near threshold values (e.g., p = 0.056) is appropriate. The decision to include only accuracy, and not sensitivity/specificity, is well-justified given the nature of the single-answer format. Findings are cautiously interpreted, and limitations—including potential biases and training data issues—are transparently discussed. Overall, the conclusions are supported by the data and responsibly presented.

Additional comments

This revised version of the manuscript is a substantial improvement over the original. The authors have taken reviewer feedback seriously, incorporated new literature, strengthened methodological transparency, and refined the clarity of their discussion. The work presents a timely and valuable contribution to the evaluation of AI chatbots in pharmacological education. I support its acceptance in its current form.